# Impacts of Protein from High-Protein Rice on Gelatinization and Retrogradation Properties in High- and Low-Amylose Reconstituted Rice Flour

**Yitong Zhao** [1,†] **, Xianggui Dai** [1,†] **, Enerand Mackon** [2] **, Yafei Ma** [1] **and Piqing Liu** [1,*]

1   State Key Laboratory of Conservation and Utilization of Subtropical Agro-Bioresources, College of Agriculture, Guangxi University, Nanning 530005, China; zhao-yitong@st.gxu.edu.cn (Y.Z.); 2017301002@st.gxu.edu.cn (X.D.); mayafei@st.gxu.edu.cn (Y.M.)

2   State Key Laboratory of Conservation and Utilization of Subtropical Agro-Bioresources, College of Life Science and Technology, Guangxi University, Nanning 530005, China; breedermackon@st.gxu.edu.cn

*   Correspondence: liupq@gxu.edu.cn; Tel.: +86-133-9770-0471

†   These authors contributed equally to this work.

**Abstract:** High-protein rice is nutritional, but its taste attributes are inferior to low-protein rice. Many documents correlate its taste attributes with its gelatinization and retrogradation properties. This study investigated the changes in gelatinization and retrogradation properties of low- and high-amylose reconstituted rice flour (RRF) added with different fractions of proteins extracted from high-protein rice. The addition of protein decreased the RVA (rapid viscosity analyzer) viscosity parameters of the RRF but increased the peak time. The high amylose fractions in the RRF mainly increased the parameters PV, FV, SB, and peak times, and scarcely affected the parameters BD and PaT. The interaction between amylose and protein determined the pasting temperature. Protein addition in RRF significantly decreased gelatinization enthalpies but increased the onset temperature ($T_o$) and peak temperature ($T_P$), while the amylose in RRF increased the gelatinization enthalpies, $T_o$ and $T_P$. Protein additions decreased the gel hardness and the pore size, while the amylose increased the gel hardness but decreased pore size. Our findings may be potentially useful in breeding and cultivating high-protein rice.

**Keywords:** reconstituted rice flour; retrogradation; gelatinization; thermal properties



## 1. Introduction

Rice feeds half of the world's population and plays a vital role in the food security of developing countries in the world. Rice proteins are of high quality, mainly because proteins from rice are gluten-free, of low allergies, and easily digestible; as a result, rice flour and its products are deeply liked by children, older people, and especially by celiac disease patients [1]. Although high-protein rice (more than 10.00% protein in milled rice) is considered superior to low-protein rice regarding its nutritional value, its mouthfeel when cooked and used in other products is considered inferior to low-protein rice [2–4]. The gelatinization and retrogradation properties of rice flour determine the rice flour products' sensory characteristics and edible quality [5]. Therefore, it is necessary to study the effects of protein from high-protein rice on gelatinization and retrogradation properties.

The parameters peak viscosity (PV), final viscosity (FV), break down (BD), set back (SB) and pasting temperature (PaT) obtained from RVA can specifically show the change in gelatinization and short-term retrogradation performance of rice flour. Several studies have shown that protein content (PC) is one of the main factors affecting the gelatinization properties of starch. The increase in protein fractions in the starch led to the decrease in PV, FV, BD, and SB, whereas PaT increased [6,7]. The globulin in the starch decreases PV, BD, FV, and PaT, while glutelin in the starch decreases PV, BD, and FV but increases PaT [8]. The effects of amylose on gelatinization vary with different studies. Previous

studies have demonstrated that the increase in amylose content (AC) in the starch will lead to the increase in PaT and the decrease in BD [5,9], while one study points out that it is amylopectin and not amylose that determines the BD value [10].

*Starch retrogradation* is when disaggregated amylose and amylopectin chains in a gelatinized starch paste reassociate to form ordered structures. The gels made from the starch with added rice protein show a low hardness compared with the control during the retrogradation [6]. The presence of protein in rice gel inhibits the migration of water and the recrystallization of starch, leading to a retardation of the retrogradation of the gelatinized starch [11]. Since amylose is the main cause of rice starch retrogradation, and starch retrogradation contributes to the changes of textural properties of cooked rice during storage, cooked rice with high amylose content is likely to become hard during storage [12].

The protein from high-protein rice may be different from the protein of normal rice. The combinations of protein and starch from different rice cultivars will inevitably lead to discrepancies in gelatinization and retrogradation performance, resulting in the different edible quality of high-protein rice. To quantify this discrepancy, we used reconstituted rice flour composed of proteins from high-protein rice and starches from two cultivars to investigate its gelatinization and retrogradation properties, which provide some clues for the quality breeding of high-protein rice.

## 2. Materials and Methods

### 2.1. Materials

In this study, two cultivars were used, namely Zhenguiai (ZGA) and 4112B. ZGA is a cultivar with high AC (25.9%) and 10.11% protein content (PC), widely used in the rice noodle industry of China. 4112B is a cultivar with low AC (13.8%) and 14.7% PC. Other chemicals such as sodium hydroxide, hydrochloric acid, and n-hexane were purchased from Yurui Technology Co., Ltd., Shenzhen, China.

### 2.2. Preparation of Rice Protein and Starch

The rice protein and starch extraction were carried out according to the procedure described by Wang et al. [6] with slight modification. Briefly, rice was ground to obtain flour and passed through 80 mesh sieves. Then, the flour was degreased by adding 1:4 (*w/v*) n-hexane. The defatted rice flour was dried for 24 h at ambient temperature, then put into NaOH solution (0.05 M) in the ratio of 1:4 (*w/v*), stirred at 500 rpm at room temperature for 2 h, then centrifuged at $6000 \times g$ for 15 min. The supernatants were collected, and the process was repeated three times. After each process, supernatants were collected. Their pH was adjusted to 4.8 (the isoelectric point) with 0.1 M HCl. Then, the proteins were precipitated from the supernatants for 1 h at room temperature. The precipitated proteins were centrifuged at $8000 \times g$ for 20 min, and the new precipitate was collected and freeze-dried. The rice starch obtained by the centrifugation of alkali liquor was washed to neutral with deionized water and freeze-dried. Protein and starch were stored at $-20\,^{\circ}$C.

### 2.3. Preparation of High-Amylose and Low-Amylose Reconstituted Rice Flour (RRF) with Different Protein Fractions

The high-amylose RRFs were prepared by mixing the starch extracted from ZGA with 0%, 6%, and 12% (*w/w*) of protein extracted from the high-protein rice 4112B. The low-amylose RRFs were prepared by mixing the starch extracted from 4112B with 0%, 6%, and 12% (*w/w*) of protein extracted from the high-protein rice 4112B.

### 2.4. Determination of RVA Spectrum of Reconstituted Rice Flour

The reconstituted rice flour was dispersed in deionized water in the ratio of 1:2 (*w/w*) [13]. After magnetic stirring at room temperature (800 rpm) for 1 h, the gelatinization characteristics of the sample were measured with a rapid viscosity analyzer (RVA4500, Perten, Stockholm, Swede). The program setting was to heat the rice slurry from room

temperature (about 20 °C) to 95 °C at the rate of 5 °C/min, then hold it at 95 °C for 4 min, cool to 50 °C at the rate of 5 °C/min, and then hold for 4 min.

### 2.5. Differential Scanning Calorimetry (DSC)

To perform the DSC, 3 mg of the reconstituted rice flour was put into a crucible and then dispersed in distilled water at a ratio of 1:2 (*w/w*), and left standing at 4 °C for 12 h. A differential heat scanner (SDT650, TA Instruments-Waters LLC, New Castle, DE, USA) was calibrated with indium before using for measurement. The sample was heated from 25 °C to 95 °C at a heating rate of 10 °C/min, and the starch was completely gelatinized. Parameters such as onset temperature ($T_o$), peak temperature ($T_p$), conclusion temperature ($T_c$), and enthalpy ($\Delta H$) can be determined from DSC heat flow curves.

### 2.6. Determination of Gel Hardness of Reconstituted Rice Gel

The determinations of the gel hardness were based on the literature [14] with modifications. The reconstituted flour (2.5 g) was dissolved in 5 g of deionized water, stirred at 800 rpm for 1 h at room temperature, then placed in a 95 °C water bath for 15 min to let the reconstituted flour gelatinize. Afterward, the gelatinized flour was stored at 4 °C for 10 min for cooling and then molded into small gel pieces (diameter, 2 cm; height, 1 cm). The hardness of the gel was determined by a texture analyzer (TMS-PRO, FTC, Sterling, VA, USA) and P6 probe. The sample was compressed to 75% of the original height at the rate of 1 mm/s, and the maximum force (g) for keeping the final height was the hardness of the gel.

### 2.7. Scanning Electron Microscopy (SEM)

The sample preparation and experimental procedures were referenced and modified from previous experiments [6,15]. The reconstituted rice flour (2.5 g) was dissolved in 5 g of deionized water, stirred at room temperature (800 rpm) for 1 h, then soaked in water at 95 °C for 15 min to make it gelatinized completely. The paste was stored at 4 °C for different periods of 1, 7, and 14 days, and the gel was observed through a scanning electron microscope (ZEISS EVO, ZEISS, UK). Before observation, the gel was pre-cooled for 24 h at −80 °C, dried through a vacuum at −40 °C, cut with a stainless-steel blade into small pieces, and adhered to the sample rack. The microscopic structure of the gel was observed at 20 kV.

### 2.8. Statistical Analysis

All measurements were repeated in three separate trials. Analysis of variance (ANOVA) was performed. Statistical analysis was performed using IBM SPSS statistics 26.

## 3. Results

### 3.1. RVA Parameters of the Reconstituted Rice Flour (RRF)

RVA parameters reflect the reconstituted rice flour's gelatinization and short-term retrogradation properties. The protein and amylose fractions impacted on the RVA parameters (Table 1).

In the same RRF, all the RVA parameters except the peak time and PaT of the reconstituted rice flour decreased when its protein fractions increased from 0% to 12% (wt%) regardless of the amylose fraction in the RRF. With the increase in PC, PV continuously decreases, maybe because the integration of protein and starch particles inhibited the swelling of starch particles during heating, reducing the viscosity of rice flour slurry. The more protein the starch–protein system contained, the more severely the expansion of the starch particles was inhibited, and the system had less viscosity. The FV decreased as the protein fractions were raised in both the high-amylose and low-amylose RRF. This decrease may be because the protein retards retrogradation [11]. During cooling, the paste tended to retrograde because of the rearrangement of the amylose. The viscosity increased with paste retrogradation, which was retarded by the increased protein fraction, lowering

the FV. The BD value also demonstrated the same trend. The breakdown value reflects the tolerance of the reconstituted rice flour paste under high temperature and high shear force. It indicates the stability of reconstituted rice paste and the damage degree of the particle during gelatinization [6]. The breakdown value can directly reflect the softness and hardness of the cooked rice: the cooked rice with a hard texture has a small breakdown value, while those with a soft texture have large breakdown values [16]. When the protein fractions of the reconstituted rice flour increased from 0% to 12%, the breakdown value of the high-amylose RRF and low-amylose RRF decreased from $543.0 \pm 19.7$ and $650.6 \pm 50.9$ to $315.3 \pm 64.5$ and $334.6 \pm 69.1$, respectively. These results indicated that added protein could mitigate the disintegration of the starch particles in the rice flour. Furthermore, the setback value of the high-amylose RRF and low-amylose RRF also decreased from $1864.2 \pm 13.8$ and $893.4 \pm 82.1$ to $1014.5 \pm 129.0$ and $596.9 \pm 86.2$, respectively, with the increase in protein fractions in the reconstituted rice flour, which showed that the protein introduction could significantly retard the short-term retrogradation of rice flour. In addition, with the increase in protein fractions, the peak time of the high- and low-amylose RRF increased slightly, from $6.00 \pm 0.04$ min to $6.27 \pm 0.00$ min and from $5.38 \pm 0.00$ min to $5.63 \pm 0.10$ min, respectively, suggesting that the integration of protein and starch particles delayed the burst of starch particles during heating. When the protein fractions were raised, the PaT increased significantly in high-amylose RRF, from $78.3 \pm 0.4$ °C to $81.8 \pm 2.1$ °C, but decreased insignificantly from $80.0 \pm 0.5$ °C to $79.5 \pm 0.5$ °C in low-amylose RRF. This phenomenon indicated that the interaction between the protein and amylose determined the pasting temperature of the RRF.

**Table 1.** Pasting properties of high- and low-amylose RRF composites with protein fractions (wt%).

| RRF | Protein Fraction | PV (cp) | BD (cp) | FV (cp) | SB (cp) | PeakTime (min) | PaT (°C) |
|---|---|---|---|---|---|---|---|
| High amylose | 0% | $1945 \pm 46.4$ Aa | $543.0 \pm 19.7$ Aa | $3266.5 \pm 67.9$ Aa | $1864.2 \pm 13.8$ Aa | $6.00 \pm 0.04$ Aa | $78.3 \pm 0.4$ Aa |
| High amylose | 6% | $1668 \pm 115.8$ Ab | $344.9 \pm 46.4$ Ab | $2930.3 \pm 247.4$ Aa | $1607.1 \pm 180.4$ Aa | $6.05 \pm 0.22$ Aa | $79.4 \pm 1.3$ Aa |
| High amylose | 12% | $1247 \pm 126.6$ Ac | $315.3 \pm 64.5$ Ab | $1946.1 \pm 187.6$ Ab | $1014.5 \pm 129.0$ Ab | $6.27 \pm 0.00$ Ab | $81.8 \pm 2.1$ Aa |
| Low amylose | 0% | $1581 \pm 90.1$ Ba | $650.6 \pm 50.9$ Aa | $1823.7 \pm 52.5$ Ba | $893.4 \pm 82.1$ Ba | $5.38 \pm 0.00$ Bb | $80.0 \pm 0.5$ Aa |
| Low amylose | 6% | $1281 \pm 142.8$ Bb | $402.2 \pm 87.8$ Ab | $1514 \pm 203.5$ Bb | $635.6 \pm 149$ Bb | $5.58 \pm 0.09$ Aa | $79.9 \pm 0.5$ Aa |
| Low amylose | 12% | $1135 \pm 78.8$ Ac | $334.6 \pm 69.1$ Ab | $1397.5 \pm 99.6$ Ab | $596.9 \pm 86.2$ Bb | $5.63 \pm 0.10$ Ba | $79.5 \pm 0.5$ Aa |
| Significance (*F*-value) | | | | | | | |
| RRF | | * | ns | * | * | * | ns |
| Protein fraction | | * | * | * | * | * | ns |
| RRF × Protein fraction | | ns | ns | * | * | ns | * |

**Note:** Mean $\pm$ standard deviation values, if followed by different lower-case letters within an amylose group (high amylose/low amylose), indicate their significant difference at $p < 0.05$; if followed by upper case letters, indicate their significant differences between the same protein fraction treatments of the different amylose RRF at $p < 0.05$. High-amylose RRF contained 25.9% amylose; low-amylose RRF contained 13.8% amylose. PV: peak viscosity; BD: breakdown; FV: final viscosity; SB: setback; PT: pasting temperature. *: Significant difference at the probability of 0.05 ($p < 0.05$). ns: no significant difference.

The high-amylose fractions in the RRF mainly increased the PV, FV, SB, and peak times, and almost did not affect the BD and PaT. When the starch suspension was heated to the PV, almost all amylose was leached out [17], so amylose influence on the PV should be limited since almost no amylose physically existed in the starch granules at that time. The rice starch, which contains a high level of amylose, usually contains a large fraction of long branch-chain amylopectin [18,19]. The longer branch-chain amylopectin the starch has, the larger the starch granules swell before they burst, and the starch has a larger PV [20]. The high-amylose fractions in the RRF increased the PV compared with the low-amylose RRF. Since the amylose can stabilize the starch granule in hot water [21], the high-amylose RRF granules should need more time to burst than the low-amylose RRF granules, increasing the PaT. The contraction of the amylose molecular chain conformation is the major factor that causes the short-term retrogradation of starch. The high-amylose RRF had larger FV and SB values compared with the low-amylose RRF because the short-term retrogradation mainly determined these two parameters. The BD values in the high-amylose RRF decreased but did not differ significantly compared to those in the low-amylose RRF at the same protein fractions, indicating that the amylose fraction was not a key determinant of the BD values.

### 3.2. Thermal Properties of Reconstituted Rice Flour

The addition of protein significantly altered the thermal properties of the reconstituted rice flour (Table 2). The $T_o$, $T_p$, and $T_c$ increased, and the $\Delta H$ decreased when we added the protein to the RRF with different AC. The denaturation temperature of the protein (86.52 °C) is higher than the gelatinization temperature of the starch, so when the protein fraction in starch rises, the endothermic peak in the system will be close to the denaturation temperature of protein [22], which might be the reason for the increase in these $T_o$, $T_p$, and $T_c$ parameters. The $\Delta H$ in DSC reflects the amount of energy required for disrupting the double helices of amylopectin in starch granules [16,23]; substituting the starch with protein in the RRF reasonably decreased the amylopectin fraction, reducing the $\Delta H$. The amylose fraction affected the thermal properties of the RRF, since the $T_o$, $T_p$, and $\Delta H$ increased as the amylose increased; the reasons behind these changes remain elusive and need to be addressed in future studies.

**Table 2.** The thermal properties of reconstituted rice flour (RRF) with different AC and protein fractions (wt%).

| RRF | Protein Fraction | $T_o$ (Onset, °C) | $T_p$ (Peak, °C) | $T_c$ (Conclusion, °C) | $\Delta H$(J/g) |
|---|---|---|---|---|---|
| High amylose | 0% | 66.50 ± 1.14 Ab | 73.43 ± 0.71 Ab | 80.01 ± 0.59 Ab | 10.66 ± 0.68 Aa |
| High amylose | 6% | 67.86 ± 0.81 Ab | 74.37 ± 1.04 Ab | 80.96 ± 0.63 Ab | 8.82 ± 1.17 Ab |
| High amylose | 12% | 69.21 ± 0.96 Aa | 76.43 ± 1.59 Aa | 82.56 ± 0.41 Aa | 6.80 ± 1.55 Ac |
| Low amylose | 0% | 65.15 ± 0.95 Bc | 71.15 ± 0.85 Bc | 79.17 ± 0.75 Ab | 8.87 ± 1.09 Ba |
| Low amylose | 6% | 66.71 ± 1.08 Bb | 73.28 ± 0.78 Bb | 80.64 ± 0.91 Ab | 7.21 ± 1.09 Bb |
| Low amylose | 12% | 68.04 ± 0.71 Ba | 74.06 ± 1.06 Ba | 82.00 ± 0.66 Aa | 5.97 ± 1.38 Bc |
| Significance (*F*-value) | | | | | |
| RRF | | * | * | ns | * |
| Protein fraction | | * | * | * | * |
| RRF × Protein fraction | | ns | ns | ns | ns |

**Note:** Mean ± standard deviation values, if followed by different lower-case letters within an amylose group (High amylose/Low amylose), indicate their significant difference at $p < 0.05$; if followed by upper case letters, they indicate their significant differences between the same protein fraction treatments of the different amylose RRF at $p < 0.05$. High-amylose RRF contained 25.9% amylose; low-amylose RRF contained 13.8% amylose. $T_o$: Onset temperature; $T_p$: Peak temperature; $T_c$: Conclusion temperature; $\Delta H$: gelatinization enthalpy; ns: no significant difference. *: Significant difference at the probability 0.05 ($p < 0.05$).

### 3.3. Texture of Reconstituted Rice Flour Gel

To obtain a deeper insight into the changes in texture induced by the retrogradation effect of the reconstituted rice flour during storage, the hardness of gels of the reconstituted rice flour after being stored at 4 °C for 1, 7, and 14 days was investigated (Figure 1). Due to the retrogradation effect of starch, the hardness of the reconstituted rice flour gels increased significantly with time of storage. The protein in the RRF gel can bind the water efficiently and reduce water migration, and the protein in the RRF gel can also act as a barrier preventing or limiting the formation of double helix in the process of retrogradation, since protein decreases the hardness of rice flour gels. In addition, protein acted as a barrier limiting the formation of double helix in retrogradation [24]. The texture of the gel from the high-amylose RRF was harder than that of the gel from low amylose RRF, suggesting that amylose is the main factor that controls retrogradation.

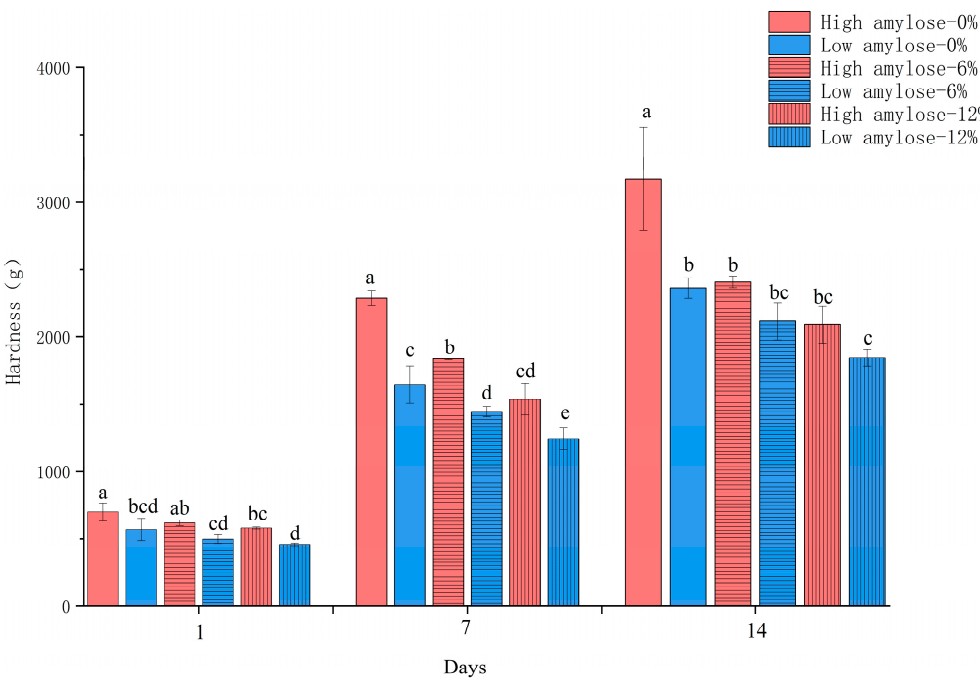

**Figure 1.** Hardness changes during high-amylose and low-amylose RRF gels storage with different protein fractions (wt%). Each column represents the mean ± standard deviation. The columns were compared on days 1, 7, and 14, and those with different letters denote where the means are significantly different ($p < 0.05$).

### 3.4. Scanning Electron Microscope (SEM)

During the storage process at low temperatures, starch retrogradation occurred, leading to changes in the microstructure of the reconstituted rice gel. The protein addition and amylose fraction affected the microstructure of the RRF gels (Figure 2). The shrinkage of the gel pores caused by retrogradation was obvious as early as 1 day after storage. As the retrogradation proceeded, the pore size became small and dense, since the protein and starch fused, causing space restriction in the starch structure [25]. The pore sizes decreased as the protein fractions in the RRF gel increased. The pore size of the RRF gel decreased when amylose increased, indicating that the starch retrogradation is a process in which disaggregated amylose and amylopectin chains in a gelatinized starch paste reassociate to form ordered structures. The more amylose the starch has, the stronger the retrogradation and the smaller the pore size will be.

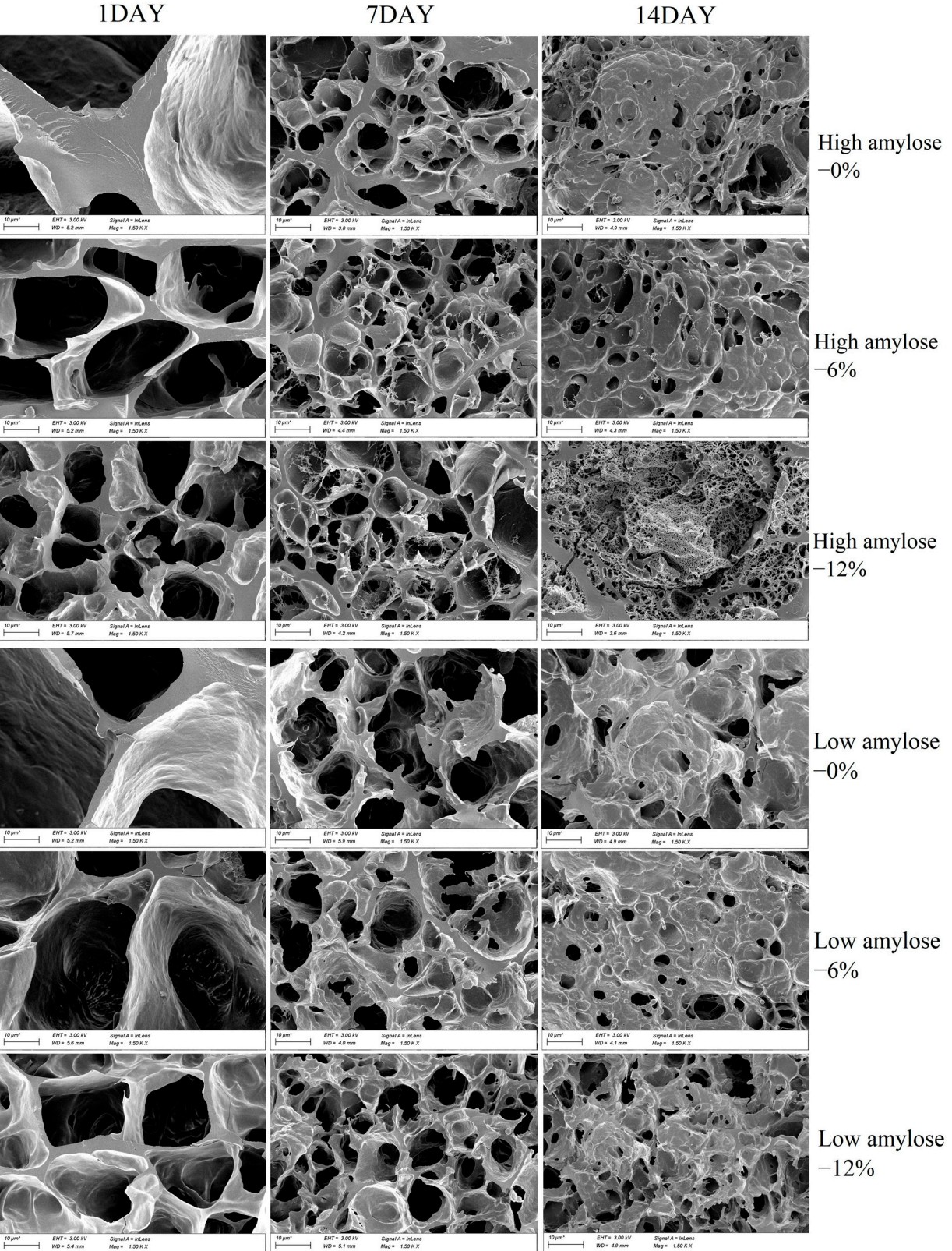

**Figure 2.** SEM images of the high-amylose and low-amylose RRF gels were added with different protein fractions (wt%). The magnification is 2000 times. High-amylose RRF contained 25.9% amylose; low-amylose RRF contained 13.8% amylose.

## 4. Discussion

The milled rice contains carbohydrates, proteins, lipids, and many other ingredients. The quantity and quality of these ingredients may affect the gelatinization and retrogradation properties in the rice flour. In our study, by using the starch–protein RRF system, we revealed that protein from high-protein rice affected the gelatinization and retrogradation properties of high- and low-amylose RRF.

Researchers have often used RVA parameters to describe the properties of flour gelatinization. In most studies, PV, BD, FV, and SB decreased when protein fractions in rice flour increased [7,9,26]; our results corroborated these previous findings. Our study showed that high-amylose RRF often had high PV, FV, and SB but low DB. Previous studies have reported similar results. By studying ten special noodle rice cultivars differing in AC, Xuan et al. [5] demonstrated that AC is positively correlated with SB and FV and is negatively correlated with BD. By studying the correlations between the pasting properties and amylose fractions, Singh et al. [27] and Noda et al. [28] also demonstrated that amylose fractions are positively correlated with PV and SB. Our study demonstrated that the interaction between protein and amylose quantities is determined by PaT, while Xuan et al. [5] demonstrated the AC positively correlated with PaT. This contradicting result may be because they used rice flour, which has various ingredients, and we used simple RRF, which only has two ingredients: starch and protein. Our study showed that protein and AC only altered PaT slightly. This result may be consistent with the previous conclusion, which suggested that the gelatinization temperature is mainly controlled by the proportion of the middle length BC amylopectin, conditioned by the locus *ALK* [29]. As for the effects of protein on $T_o$, $T_p$, and $\Delta H$, the protein addition to rice starch raises $T_o$ and $T_p$ and decreases $\Delta H$ [30,31], which is consistent with our results. In the case of amylose, $T_o$, $T_p$, and $\Delta H$ can be increased by increasing the AC, and similar results were documented in the previous findings [12,32].

The protein and amylose quantity can alter the retrogradation properties of the starch gel or cooked rice. The addition of protein or proteolysates to the starch gel will reduce the hardness of the retrogradation gel [31,33]. A high amylose content can increase the hardness of the starch gel [34] and the cooked rice [12]. Our results support these previous findings.

Since the RVA parameters and the retrogradation parameters have been associated with the mouthfeel of cooked rice, our present study intends to unravel the secrets surrounding the mouthfeel of high-protein rice by investigating the effects of protein from high-protein rice on gelatinization and retrogradation properties. A high BD, a low SB, and a low PaT correlate positively with a good mouthfeel of cooked rice [35,36]. Our study demonstrated that as the protein fractions in the RRF increased, the parameter BD decreased significantly ($p < 0.05$). Maybe the low BD in high-protein rice is one reason that makes high-protein rice of inferior eating quality. As the amylose fraction in RRF decreased, the parameter BD increased, but not significantly. This finding implies that when breeding high-protein rice with a good mouthfeel, high-protein rice should have low amylose in starch. The protein composition also affects the taste of the rice. In *japonica* rice, the ratio of glutelin/prolamin and the globulin content are found to be significantly and positively correlated with the taste value [37], which indicates that, in the breeding of high-protein rice with good taste, the ratio of glutelin/prolamin and the globulin content should be taken into account. Retrogradation occurs in the precooling stage of cooked rice and causes the cooked rice to harden, and softness is an indicator of the rice's good palatability [38]. Our study suggests that low-amylose RRF gel is soft, and the previous study shows that the AC of cooked rice positively correlates with the hardness of cooked rice during storage [12]. Previous studies [6] and our present study demonstrate that the protein added in starch softens the starch gel and cooked rice; therefore, the protein from high-protein rice contributes to the good palatability of the cooked rice in terms of the hardness. Our study revealed that breeding high-protein rice with a good taste is possible if we can combine the high-protein trait, low-gelatinization-temperature trait, and low-amylose trait in one cultivar.



## 5. Conclusions

The protein from high-protein rice significantly affected the gelatinization and retrogradation properties of high- and low-amylose RRF. The addition of protein decreased the RVA viscosity parameters of the RRF but increased the peak time. The high amylose fractions in the RRF mainly increased PV, FV, SB, and peak times and scarcely affected the BD and PaT. The interaction between amylose and protein determined and altered the pasting temperature slightly. The protein from the high-protein rice slowed down the retrogradation of the RRF gel, while the amylose in RRF promoted the retrogradation of the RRF gel. Our findings suggest that to breed high-protein rice with a good taste, the breeder should combine the low-amylose gene, low-gelatinization-temperature gene, and the high-protein gene into one cultivar.

**Author Contributions:** Conceptualization, investigation, formal analysis, writing—original draft preparation Y.Z., X.D.; writing—review, E.M., Y.M.; funding acquisition, conceptualization, writing—review and editing, supervision, P.L. All authors have read and agreed to the published version of the manuscript.

**Funding:** This work and APC were funded by the national R&D Priority Program—Breeding New Rice Varieties for Southern China Area (2017YFD0100100) and the Guangxi R&D Priority Program—Research and application of rice resistance breeding to bacterial blight (Guike AB 16380124).

**Institutional Review Board Statement:** Not applicable.

**Informed Consent Statement:** Not applicable.

**Data Availability Statement:** Raw data can be provided to researchers on request by corresponding with the first author.

**Acknowledgments:** We are thankful to Fei Li for his support in the lab experiments.

**Conflicts of Interest:** The authors declare no conflict of interest.

## Abbreviations

| Abbreviation | Description |
| --- | --- |
| AC | Amylose Content |
| BD | Breakdown |
| DSC | Differential scanning calorimetry |
| FV | Final Viscosity |
| PaT | Pasting Temperature |
| PV | Peak Viscosity |
| RRF | Reconstituted Rice Flour |
| RVA | Rapid Viscosity Analyzer |
| SB | Set Back |
| SEM | Scanning electron microscope |
| ZGA | Zhenguiai |

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
