# Peer review of "Impacts of Protein from High-Protein Rice on Gelatinization and Retrogradation Properties in High- and Low-Amylose Reconstituted Rice Flour"

_agronomy, doi:10.3390/agronomy12061431_

Round 1

Reviewer 1 Report

The author has resubmitted with sufficient supplements and revisions to what I had previously suggested. Therefore, I suggest that this manuscript would be accepted without major revisions.

Author Response

Thanks very much for taking your time to review this manuscript. I really appreciate all your work!

Reviewer 2 Report

First of all I would like to apologizer a overdue review.

This manuscript, in my opinion, will rerquire only a few corrections. I have some dous about applied statistical evaluation method, but the existing one still is "working"

Author Response

Dear reviewer:

  Thanks very much for taking your time to review this manuscript. I really appreciate all your work. 

Kind regards

Reviewer 3 Report

Authors used extracted protein to make reconstituted rice flour, and compared how protein and amylose effect gelatinization pasting and texture properties. The experiment is well designed. However, data interpretation and mechanism discussion need more work. Some sentences are self-contradicted.  

Line 47. ‘The effects on gelatinization of the amylose in the starch can vary with different studies.’ After reading this, I suppose author it is going to talk about protein effects on the amylose gelatinization. However, it is followed by amylose content effects on starch gelatinization. Revised it to make more cohesive. 

 Line 57-59. Two non-related sentences are put in one sentence. Need re-write it. 

Line 60. What is ‘the protein of the ordinary protein’? 

Line 28. In the Introduction part, authors have not given a convincing reason why this research project need to be done, what potential benefits can people get from results? High protein content rice can be healthy and also tasty? 

Line 67. How could results finally help breeding of high protein rice

Line 88. The amylose content is contradictory to amylose content listed in Line 72.  

Line 92-95. For RRF, why author did not use protein extracted from ZGA, but only protein from 4112B?

Line109. ‘TP’, ‘p’ is subscript?

Line114. During gelation stage, mixing was applied ?

Line 127. ‘dried through vacuum’, at which temperature? or was freeze dried?

Line 141. What does ‘fluctuation’ mean? What are the comparing samples? In the table 1, with the increase of protein content, PV continuously decreases? 

Line 145-147. All viscosity parameters decrease with the increase of protein content. Final viscosity decreases may result from starch did not expand enough in the early gelation stage, causing lower viscosity data in the consequent measure (including BD and FV). Authors believe it is due to the lack of retrogradation, which is arguable. 

Line 171-172. ANOVA analysis shows there is no significant difference, how conclusion ‘This 170 phenomenon indicated that the interaction between the protein and amylose determined 171 the pasting temperature of the RRF.’ was made?

Line203-204. Authors need to explain how protein denaturation temperature (86.52) increase DSC temperatures?

Line 207-208. Enthalpy (H) decreases simply because of less starch content (replaced by protein)? 

Line 211. In line 206, author stated that Enthalpy measures the amount of amylopectin double helices, so starch with less amylose should have higher enthalpy data. But why in Table 2, low amylose starch shows lower enthalpy?

Line 219. ‘the addition of protein decreased the hardness of rice flour gels ’is that possible protein acts as a barrier to prevent/limit the double helixes formation during the retrogradation? 

Line 241. What causes gel pores? After drying, water was replaced by air?

Line 243.In the texture results, authors say adding protein can reduce hardness. But in SEM figures, sample with more protein shows denser and less pore structure. Authors need to explain why, because they are contradictory.  

Line 243. Authors say ‘protein may fill the gel pore’, which is not 100% correct. The pores are filled with air after drying. Mixed flour were hearted at 95 degree, so there are no protein granules/structure anymore. Protein and starch were all gelatinized and entangled with each other. Advise to use Confocal Microscopy to visualize the interaction between starch and protein. 

Author Response

Comments and Suggestions for Authors#3

General comments from REV #: Authors used extracted protein to make reconstituted rice flour, and compared how protein and amylose effect gelatinization pasting and texture properties. The experiment is well designed. However, data interpretation and mechanism discussion need more work. Some sentences are self-contradicted.  

General reply: We have made some revisions in the text, based on the valuable suggestions and comments from the REV#2.

  1. Line 47. ‘The effects on gelatinization of the amylose in the starch can vary with different studies.’ After reading this, I suppose author it is going to talk about protein effects on the amylose gelatinization. However, it is followed by amylose content effects on starch gelatinization. Revised it to make more cohesive. (Line49)

Answer: Thank the reviewer for the valuable suggestion. We have already revised the text to make it clear and cohesive.

  1. Line 57-59. Two non-related sentences are put in one sentence. Need re-write it. 

Answer: Thank the reviewer for the valuable suggestion. We have already rewritten it. (Line60)

  1. Line 60. What is ‘the protein of the ordinary protein’? 

Answer: What we mean is the protein from normal rice, we have already revised it. (Line66)

  1. Line 28. In the Introduction part, authors have not given a convincing reason why this research project need to be done, what potential benefits can people get from results? High protein content rice can be healthy and also tasty? 

Answer: Thank the reviewer for the valuable suggestion for Line 28. We revised the introduction to highlight the reasons for the study.

The reasons for the research can be summarized as follow:

  • High protein rice is important.
  • Although the high protein rice (more than 10.00% protein in the milled rice) is considered to be superior to the low protein rice in regard of its nutritional value, its mouthfeel of cooked rice and rice products is regarded to be inferior to other low protein rice.
  • The gelatinization and retrogradation properties of rice flour determine the sensory characteristics and edible quality of rice flour products.

The benefits that people can get from this result are mainly discussed in the discussion section:

  • Unraveling the reason for the low eating quality for the high protein rice: the low BD in the high protein rice may be one of the reasons that make the high protein rice of the inferior eating quality.
  • In order to breed the high protein rice with the good mouthfeel, the high protein rice should have the low amylose in the starch.
  1. Line 67. How could results finally help breeding of high protein rice?

Answer: Thank the reviewer for the question. The gelatinization and retrogradation parameters can be used to guide the breeding of the quality high protein rice. The gelatinization parameter--gelatinization temperature is widely used in the evaluation of the quality rice in China. In the future, more parameters in the gelatinization and retrogradation will be used to unravel the secrets of the rice taste, and to facilitate the breeding of the quality rice.

  1. Line 88. The amylose content is contradictory to amylose content listed in Line 72.  

Answer: Thanks reviewer’s valuable comment. We made a mistake, and corrected it. (Line77)

  1. Line 92-95. For RRF, why author did not use protein extracted from ZGA, but only protein from 4112B?

Answer: Thanks reviewer’s valuable suggestion. Because the purpose of the study was to investigate the effects of the protein from the high protein rice on the gelatinization and retrogradation of the high amylose and low amylose reconstituted rice flour (RRF), we only used the protein from 4112B.

  1. ‘TP’, ‘p’ is subscript?

Thank the reviewer for the comment. Tp is correct, and the text has been revised. (Line116)

  1. During gelation stage, mixing was applied?

Answer: Thank reviewer for the question. Yes, during gelation stage, mixing was applied.

  1. Line 127. ‘dried through vacuum’, at which temperature? or was freeze dried?

Answer: Thank the reviewer for the valuable question. We use vacuum freeze-drying, temperature (-40℃), revised.

  1. Line 141. What does ‘fluctuation’ mean? What are the comparing samples? In the table 1, with the increase of protein content, PV continuously decreases? 

Answer: Thanks reviewer’s valuable suggestion. The word ‘fluctuation’ is not properly used here. We revised the text. (Line148)

  1. Line 145-147. All viscosity parameters decrease with the increase of protein content. Final viscosity decreases may result from starch did not expand enough in the early gelation stage, causing lower viscosity data in the consequent measure (including BD and FV). Authors believe it is due to the lack of retrogradation, which is arguable. 

Answer: Thank the reviewer for the comment. The explanations are based on the ref 11. In fact, we did not have the evidence to prove the final viscosity decrease was due to the lack of retrogradation in this study, which needs to be addressed in the future study. That is the reason we used the word ‘may’ in the text.

  1. Line 171-172. ANOVA analysis shows there is no significant difference, how conclusion ‘This phenomenon indicated that the interaction between the protein and amylose determined the pasting temperature of the RRF.’ was made?

Answer: Thank reviewer for the question. ANOVA analysis shows there is no significant difference for RRF and protein fraction, but there is significant difference for RRF and protein fraction interaction (Table 1). Hereafter we got conclusion ‘This phenomenon indicated that the interaction between the protein and amylose determined the pasting temperature of the RRF’ (Line 171-172).

  1. Line203-204. Authors need to explain how protein denaturation temperature (86.52) increases DSC temperatures?

Answer: Thank the reviewer for the valuable suggestion. The denaturation temperature of protein is higher than the starch gelatinization temperature. When the fraction of protein in the system increases, the thermal stability of the system is improved, so the system will increase the DSC temperatures.

  1. Line 207-208. Enthalpy (△H) decreases simply because of less starch content (replaced by protein)?

Answer: Thank reviewer for the question. The experimental data show if we decrease the ratio of the starch : protein, the enthalpy (△H) decreases. This result is consistent with the previous result [1], which explained that the decrease of gelatinization enthalpy may be due to the decrease of starch ratio.

[1] Wang, L.; Zhang, L.; Wang, H.; Ai, L.; Xiong, W. Insight into protein-starch ratio on the gelatinization and retrogradation characteristics of reconstituted rice flour. Int. J. Biol. Macromol. 2020, 146, 524-529.

  1. Line 211. In line 206, author stated that Enthalpy measures the amount of amylopectin double helices, so starch with less amylose should have higher enthalpy data. But why in Table 2, low amylose starch shows lower enthalpy?

Answer: Thank the reviewer for this good question. The theory that enthalpy measures the amount of amylopectin double helices can explain well the high enthalpy in the high protein RRF, but cannot explain the high enthalpy in the high amylose RRF. We do not know the reasons, maybe because the structure of the amylopectin in the high amylose RRF is different from that of the amylopectin in low amylose RRF, the amylopectin amount in the high amylose RRF is not proportional to the amount of the double helices, and the amylopectin in the high amylose RRF may have more double helices than the amylopectin in the low amylose RRF. The above assumption needs to be addressed in the future study.

  1. Line 219. ‘the addition of protein decreased the hardness of rice flour gels ’is that possible protein acts as a barrier to prevent/limit the double helixes formation during the retrogradation? 

Answer: Thank you for your good suggestion. Your suggestion is another assumption, and has been added in the article.

  1. Line 241. What causes gel pores? After drying, water was replaced by air?

Answer: Thank you for your work. We think you are right. After the freeze-drying of the gel, some water is replaced by air, creating pores in the gel.

  1. Line 243.In the texture results, authors say adding protein can reduce hardness. But in SEM figures, sample with more protein shows denser and less pore structure. Authors need to explain why, because they are contradictory.  

Answer: Thank reviewer for this question. In SEM figure, the pores in the gel with more proteins added are denser and more homogeneous than the pores in the gel with fewer proteins. This structure can make the high protein gels have high water holding capacity, and soft [4].

[4]    Zhang, Y.; Chen, C.; Chen, Y.; Chen, Y. Effect of rice protein on the water mobility, water migration and microstructure of rice starch during retrogradation. Food Hydrocolloid. 2019, 91, 136-142.

  1. Line 243. Authors say ‘protein may fill the gel pore’, which is not 100% correct. The pores are filled with air after drying. Mixed flour were hearted at 95 degree, so there are no protein granules/structure anymore. Protein and starch were all gelatinized and entangled with each other. Advise to use Confocal Microscopy to visualize the interaction between starch and protein. 

Answer: Thank you very much for your comment. Our explanation is based on the previous reference. The mixed flour was heated at 95 degree for 15 minutes, and the heat may destroy the secondary structure or tertiary structure of the protein, and ‘denature’ the protein, usually does not destroy the primary structure of the protein. Anyway there may be some proteins or peptides that are not degraded, which may fill the pores of the gel. Your comment is right. We should use Confocal Microscopy to verify our assumption in the future study.

Thanks very much for taking your time to review this manuscript. I really appreciate all your work. 

Kind regards

Yitong Zhao

Corresponding author: Piqing Liu

E-mail address: [email protected]

Reviewer 4 Report

The paper is quite well done, with sufficient info in each section

Author Response

Dear reviewer:

  Thanks very much for taking your time to review this manuscript. I really appreciate all your work. 

Kind regards

This manuscript is a resubmission of an earlier submission. The following is a list of the peer review reports and author responses from that submission.

Round 1

Reviewer 1 Report

The manuscript entitled "Impacts of Protein from the High Protein Rice on Gelatinization and Retrogradation in the High and Low Amylose Reconstituted Rice Flour" studied the changes in gelatinization and retrogradation properties of the low and high amylose reconstituted rice flour (RRF) containing various ratios of proteins extracted from the high protein rice. The paper presents some interesting results concerning the effects of protein on viscosity parameters and their relationship with amylose content. The explanation of the phenomena observed is quite satisfactory whereas the text is fairly well written and exhibits an element of originality.

Reviewer 2 Report

This study was discueed the relation between protein from high protein rice and amylose reconstituted rice flour, my comments as follows:

1.The author sholud improve the english writing in all contex of this manuscript extensively.

2.There are lots of logistic problems in the contex of this manuscript, " high protein rice is nutritional, but its taste attributes are inferior to the low propein rice."?

3.The tilte "impacts of protein from the high protein rice?" would made confused to the reader.

4.High and low amylose reconstituted rice flour?

5.Why the author measure the 7 and 14 gel?

6.Protein is not the only factor would affect the gelatinization and retrogradation of rice flour, the author should also discussed the effect of starch and lipid.

7.In DSC, the author should alos present the results of onset temperature (To) and Tc, and the Tp of DSC was differ to the PaT of RVA.

Reviewer 3 Report

It was interesting results in the starch-protein system, however, there are so many problems with language writing, grammar, experimental methods, and discussion etc.  I decide to reject this article  and There are some comments and suggestions to be revised as follows. 

1) Line 71; revise ‘widely used in China rice noodle industry’ to 'widely used in rice noodle industry of China'.

2) Line 77; revise ‘Wang et al. (2020)’ to ‘Wang et al. [Ref#]’

3) Line 77; revise ‘grind’ to ‘ground’

4) Line 83; Line 83; what is the purpose of adjusting the pH to 4.8?

5) Line 92; Authors prepared the sample by mixing starch and protein extracted from rice flour. It is a starch-protein mixture, not rice flour.

6) Line 98; revise ‘50 °Cf or 4min’.

7) Line 112; how long is the store at 4 °C?

8) Line 115; please rewrite more clearly.

9) Line 139; I don't understand this sentence.

10) Line 144; revise reference as ‘gelatinization [6]’.

11) Line 180-183; The sentence is written opposite to the result.

12) The contents of Figure 1 and Table 1 are duplicated. The results of Figure 1 can be explained by additional statistical processing in Table 1.

13) Line 188; generally, thermal properties are identified by measuring onset temperature, peak temperature, conclusion temperature and gelatinization enthalpy. Why did you measure only peak temperature and gelatinization enthalpy?

14) Line 193-194; this sentence is not correct. Amylose is a single helical polysaccharide and amylopectin is a double-helical polysaccharide. Gelatinization enthalpy in DSC was related to the melting of the double-helical structure of amylopectin.

15) Table 2.; Unify the decimal point in the data with the standard deviation.

16) Table 2.; The footnote is incorrect. The authors did not perform statistical processing on the same cultivar.

17) Figure 2.; It is evident that the hardness of the gel increases with longer storage periods. Unlike other analysis results, here the authors dealt with statistics between all samples. This is not suitable.

18) Line 225-226; Why is that? Does the protein move into a pore during storage? Further explanation is needed.

19) Figure 3.; In previous results, the authors placed the ZGA first, followed by the 4112B results, however, However, the order of results in figure 3 is reversed.

20) Line 247; revise ‘Xuan et al.’ to ‘Xuan et al.[Ref#]

21) Line 267; Mouthfeel is more appropriate than taste.

22) Line 277-278; Please let me know the level of correlation it is.

23) Line 288-289; It is a leap forward to link the results of research on a mixture of rice starch and rice protein to breeding of rice.

24) reference; The abbreviation of many journals was misrepresented